# Post-Quantum Security: Opportunities and Challenges

**DOI:** 10.3390/s23218744

**Published:** 2023-10-26

**Authors:** Silong Li, Yuxiang Chen, Lin Chen, Jing Liao, Chanchan Kuang, Kuanching Li, Wei Liang, Naixue Xiong

**Affiliations:** School of Computer Science and Engineering, Hunan University of Science and Technology, Xiangtan 411201, China; a1742701860@gmail.com (S.L.); chenyuxiang@hnust.edu.cn (Y.C.); linchen@mail.hnust.edu.cn (L.C.); lyuyu1983@hotmail.com (J.L.); cc19532661632@163.com (C.K.); aliric@hnust.edu.cn (K.L.); wliang@hnust.edu.cn (W.L.)

**Keywords:** cryptography, post-quantum encryption, quantum computing, CRYSTALS-KYBER, quantum security

## Abstract

Cryptography is very essential in our daily life, not only for confidentiality of information, but also for information integrity verification, non-repudiation, authentication, and other aspects. In modern society, cryptography is widely used; everything from personal life to national security is inseparable from it. With the emergence of quantum computing, traditional encryption methods are at risk of being cracked. People are beginning to explore methods for defending against quantum computer attacks. Among the methods currently developed, quantum key distribution is a technology that uses the principles of quantum mechanics to distribute keys. Post-quantum encryption algorithms are encryption methods that rely on mathematical challenges that quantum computers cannot solve quickly to ensure security. In this study, an integrated review of post-quantum encryption algorithms is conducted from the perspective of traditional cryptography. First, the concept and development background of post-quantum encryption are introduced. Then, the post-quantum encryption algorithm Kyber is studied. Finally, the achievements, difficulties and outstanding problems in this emerging field are summarized, and some predictions for the future are made.

## 1. Introduction

With the increasing application of networks in daily life, networks play a crucial role in various fields. Consequently, increasing emphasis has been directed towards network security. With the advancement of technology, sensors are finding increasingly widespread applications in daily life. They collect various types of data to enhance the quality of life, improve efficiency, and address various challenges. Consequently, ensuring the security of data within sensors has become crucial. The adoption of post-quantum encryption algorithms to replace traditional encryption methods is particularly important in this regard [1]. This transition helps protect sensitive information from potential threats in an era of rapid technological development. Buchmann et al. in [2] point out that public-key cryptography has become a major enabler of network resilience since its invention in the late 1970s. For example, SSL/TLS employs public-key cryptography to ensure the confidentiality and integrity of communications while using digital certificates to authenticate the identities of the communicating parties [3]; digital signature techniques use public-key cryptography algorithms to generate a digital signature associated with a document or communication that verifies the integrity and origin of the document and ensures that it has not been tampered with. And the safety of these schemes relies on the difficulty of computational problems such as factorization [4,5], discrete logarithm [6,7], and the Pell equation [8,9].

However, with the advent of quantum computing, traditional cryptography is at risk of being broken. Peter Shor, an American mathematician, proposed Shor’s algorithm in 1994 [10], which is a well-known quantum algorithm that utilizes the properties of quantum computing, such as quantum parallelism and quantum Fourier transform, to accelerate the factorization problem at an exponential rate, and the algorithm resolves the challenges of integer factorization and discrete logarithms linked to public-key cryptography. Lov Kumar Grover, an Indian computer scientist, published the algorithm in 1996 [11] for fast searching of unordered databases on quantum computers, which exploits the properties of quantum parallelism and interference by “inverting” and “reflecting” databases using quantum gate operations to quickly converge to the target element in the quantum search space. Daniel J. Bernstein and Tanja Lange, in [12], summarize the influence of Shor’s and Grover’s algorithms on traditional ciphers, as shown in Table 1.

Among them, the safety of the GMAC algorithm is based on the confidentiality and randomness of the key, and the safety of the Poly1305 algorithm is mainly based on the underlying cryptographic operations and hash functions, neither of which involves the factor decomposition problem of Shor’s algorithm or the database search problem of Grover’s algorithm, so the security of these two algorithms is not affected by Shor’s algorithm or Grover’s algorithm.

Meanwhile, from [13], it can be seen that considerable technological progress has been made in designing quantum computers. Thus, Fábio Borges et al. [14] point out that in this emerging quantum era, safer alternatives need to be investigated.

**Table 1 sensors-23-08744-t001:** Impact of Shor’s and Grover’s algorithms on traditional encryptions (security level b means that the best attack uses 2b operations) [12].

Name	Pre-Quantum Security Level	Function	Post-Quantum Security Level	Impact
**Symmetric cryptography**				
AES-128 [15]	128	Block cipher	64	Cracked by Grover’s algorithm
AES-256 [15]	256	Block cipher	128	Cracked by Grover’s algorithm
Salsa20 [16]	256	Stream cipher	128	Cracked by Grover’s algorithm
GMAC [17]	128	MAC	128	No impact
Poly1305 [18]	128	MAC	128	No impact
SHA-256 [19]	256	Hash function	128	Cracked by Grover’s algorithm
SHA-3 [20]	256	Hash function	128	Cracked by Grover’s algorithm
**Public-key cryptography**				
RSA-3072 [21]	128	Encryption	Broken	Cracked by Shor’s algorithm
RSA-3072 [21]	128	Signature	Broken	Cracked by Shor’s algorithm
DH-3072 [22]	128	Key exchange	Broken	Cracked by Shor’s algorithm
DSA-3072 [23,24]	128	Signature	Broken	Cracked by Shor’s algorithm
256-bit ECDH [25,26,27]	128	Key exchange	Broken	Cracked by Shor’s algorithm
256-bit ECDSA [28,29]	128	Signature	Broken	Cracked by Shor’s algorithm

Along with the rapid development of the Internet of Things, sensors are more and more widely used in daily life. Gradually, people begin to pay attention to the data privacy and security concerns related to IoT, so the security of sensor data is beginning to be taken seriously. Nowadays, with the rapid development of quantum computers, post-quantum encryption algorithms have become the future development trend in cryptography, and in order to safeguard the security of sensor data, the application of post-quantum encryption algorithms to sensors will dramatically improve the security of sensors.

In fact, the cryptographic community has long been working on cryptographic algorithms that can resist attacks from quantum computers, going back as far as the McEliece encryption in 1978 and 1979 [30], Merkle hash signatures [31], etc. But at that time, the threat of quantum computers to cryptographic algorithms was not clear, and there was no concept of post-quantum. It was only later that Daniel J. PQCrypto 2006: International Workshop on Post-Quantum Cryptography was organized by Professor Daniel J. Bernstein in 2006. It was an international workshop focusing on the field of post-quantum cryptography and the first international conference on anti-quantum cryptography, marking the initial development of the field of post-quantum cryptography, laying the groundwork for research in the field of post-quantum cryptography, and providing a platform for cryptography researchers to discuss and share ideas and methodologies on resistance to threats from quantum computation [32].

In 2012, the National Institute of Standards and Technology (NIST) commenced research into post-quantum cryptography. In February 2016, a global initiative was launched, inviting contributions to establish post-quantum cryptography standards. As of 30 November 2017, NIST received a total of 82 draft algorithms, and after an initial screening process, NIST published 69 drafts with algorithms constructed from four main mathematical methods:Lattice-based algorithms are one of the most encouraging post-quantum cryptographic algorithms [33]. The algorithm has significantly improved computational speed and security intensity, and the channel overhead has only slightly increased [34]. The safety of lattice-based algorithms depends on the complexity of solving problems in the lattice and can achieve various existing cryptographic constructions, such as digital signature, key exchange, encryption, attribute encryption, function encryption, and all homomorphic encryptions. Two-dimensional lattices and two different sets of lattice bases are shown in Figure 1.

2.Code-based algorithms utilize error correction codes to correct and compute the randomness errors added; a famous code-based encryption algorithm is McEliece [35]. The core idea of McEliece’s scheme is to utilize the error-correcting ability of linear codes (or error-correcting codes) to construct encryption algorithms. According to different classification criteria, the specific classifications are shown in Table 2. This cryptography method relies on a public-key infrastructure, where the public key consists of a sparse random matrix and a matrix that generates a linear code. Meanwhile, the private key includes the matrix inverse of the one used for creating the linear code.

3.Algorithms based on multivariate polynomials utilize sets of quadratic polynomials with multiple variables over finite fields to construct encryption, signature, key exchange, and related methods [36]. The safety of multivariate cryptography relies on the difficulty of solving systems of nonlinear equations, specifically the challenge of addressing multivariate quadratic polynomial problems.4.A hash-based signature algorithm was proposed by Ralph Merkel and is considered to be one of the feasible alternatives to traditional digital signatures (RSA, DSA, ECDSA, etc.) [37]. Due to the lack of effective quantum algorithms to quickly find collisions in hash functions, hash-based constructions (with sufficient output length) can resist quantum computer attacks. The Merkle tree is shown in Figure 2.

The advantages and disadvantages of the four post-quantum ciphers are shown in Table 3. Post-quantum cryptography is becoming more and more known as a new technology that will gradually replace current public-key cryptographic algorithms such as RSA, Diffie–Hellman [38], and elliptic curves [39] in the next 5–10 years.

Apart from the four common types mentioned above, superelliptic curves are also a cryptographic variant of elliptic curves widely employed in the field of post-quantum cryptography. They are characterized by their ability to be described through Weierstrass points and possess more complex geometric properties compared to non-superelliptic curves, particularly in the context of constructing secure cryptographic schemes. The intricacies of superelliptic curves and their mathematical properties grant them a high degree of security against traditional computing and potential quantum computer attacks, making them a subject of significant interest in post-quantum cryptography [40].

Finally, at the end of 2022, NIST identified the Kyber algorithm as a standardized algorithm [41]. NIST has made the Kyber algorithm a standardized algorithm in [41] and states that Kyber was chosen as the standard for post-quantum cryptographic algorithms because, in addition to its good security, its hardware and software implementations on multiple platforms achieve good performance and can be well embedded in most existing Internet protocols and cryptographic algorithm applications.

After four rounds of screening, as shown in Figure 3.

The Kyber algorithm is a key encapsulation mechanism with IND-CCA2 safety, where its safety is rooted in the hardness of solving learning with error problems based on lattice modules. The Kyber algorithm provides various parameter configurations for different safety levels, where Kyber-512 is designed to provide safety comparable to AES-128, Kyber-768 is designed to provide safety comparable to AES-192, and Kyber-1024 is designed to provide safety comparable to AES-256. Kyber-768 aims to provide safety comparable to AES-192, and Kyber-1024 aims to provide safety comparable to AES-256. The four candidate encryption algorithms currently identified by NIST can be categorized into two groups: public-key encryption and key establishment algorithms, with only the CRYSTALS-KYBER algorithm and digital signature algorithms, which contain three algorithms—CRYSTALS-DILITHIUM, FALCON, and SPHINCS+. Because the Kyber algorithm serves as the only identified public-key encryption and key establishment algorithm, in this paper, we choose to introduce the Kyber algorithm.

Kyber was established by NIST in 2022 as a standardized algorithm for post-quantum encryption, representing a new generation of security solutions. Understanding the Kyber algorithm helps people understand the importance of post-quantum cryptography and how to protect sensitive information from quantum computer threats in the future, as well as the field of post-quantum cryptography as a future direction. Most of the previous reviews describe the classification of post-quantum encryption and do not describe the standardized algorithms established by NIST in detail. In this paper, after reading a large amount of literature, Kyber is introduced in detail, and how others have implemented Kyber is described. By reading this review, one can have a deeper understanding of the concept of post-quantum encryption algorithms and their importance in the future, and one can deeply understand the construction principle of Kyber and the future direction of post-quantum encryption.

There are six sections to this paper. The Section 1 provides an introduction to post-quantum encryption. The Section 2 elaborates on the development of sensors and post-quantum encryption algorithms. The Section 3 introduces the implementation of the Kyber algorithm. The Section 4 provides an overview of the current software and hardware implementations of the Kyber algorithm. The Section 5 discusses the future development of post-quantum encryption algorithms, with one subsection focusing on the integration of Kyber with sensors and another subsection outlining the opportunities and challenges faced by post-quantum encryption. The Section 6, in the sixth part, offers a summary.

## 2. Background

Cryptography is the study of secure communication techniques that protect against potential attacks and is the science of encryption and decryption techniques, and, as such, involves the use of algorithms and mathematical methods to guarantee the safety and confidentiality of data during conveying and retention. Cryptography is primarily used to protect sensitive information, such as electronic payments, Internet communications, e-mail, and data stored in computer systems [42]. The most basic cryptographic classification is the division of message encryption into Symmetric Cryptography and Public-Key Cryptography Asymmetric Cryptography.

Symmetric encryption

Symmetric encryption is a cryptographic mechanism that employs the same key for both data encryption and decryption processes. A common key is shared between both the sender and the recipient, which the sender uses to convert the data into ciphertext, and the recipient employs the identical key to transform the ciphertext back into its original plaintext form [43]. Symmetric encryption is characterized by high speed and simple encryption and decryption processes. Conventional symmetric encryption algorithms include DES (Data Encryption Standard) and AES (Advanced Encryption Standard). Its disadvantage is the complexity of key management in a distributed environment. 

2.Asymmetric encryption

Asymmetric encryption came into existence to address the complexities of symmetric encryption in distributed key management. Asymmetric encryption is an encryption mechanism that uses different keys for encrypting and decrypting data. One of them is called a public key for data encryption and can be used by anyone; the other, known as a private key, is used to unscramble information, and only the possessor of the private key has the capability to decipher information that has been encrypted using the public key. Common asymmetric encryption algorithms include RSA (developed by Rivest, Shamir, and Adleman), ECC (Elliptic Curve Cryptography), and DSA. The complexity of mathematical problems is a guarantee of the security of these algorithms, which makes it extremely difficult to calculate the private key from the public key. Asymmetric encryption algorithms are characterized by the intricacy of the algorithms, while safety is significantly improved in contrast to symmetric encryption by [44].

RSA is very representative of traditional asymmetric encryption algorithms; the next section will be a brief introduction to RSA. The foundation of the RSA algorithm relies on a straightforward principle from number theory: multiplying two large prime numbers is a straightforward task, yet factoring their resulting product proves to be exceedingly challenging. This property enables the product to be utilized as the encryption key, specifically the public key. The two arrays of large prime numbers are amalgamated to form the private key, facilitating encryption [45]. The RSA algorithm key generation process is shown in Figure 4. The RSA algorithm is described as follows:Two primes, P and Q, are chosen at random.Find the value M of the Euler function for N. The Euler function is given by the following formula, where $p_{1}$, $p_{2}$, $p_{3}$,…, $p_{k}$ is all unrepeated prime factors of n.



(1)
$$\varphi (n)=n(1-\frac{1}{p_{1}})(1-\frac{1}{p_{2}})(1-\frac{1}{p_{3}})\ldots (1-\frac{1}{p_{k}})$$



3.Find an integer *E* that is prime to *M*.4.Find an integer *D* that satisfies the following relation:


(2)
(E×D)modM=1


To summarize, *N*, *M*, *E*, and *D* are computed by randomly choosing P and Q, which are prime to each other. Then, we divide these numbers into two groups, (*E*, *N*) and (*D*, *N*), the public key group and the private key group, respectively. *E* is the public key, and *D* is the private key.

In addition to using traditional encryption algorithms, we also frequently employ steganography for data protection in our everyday lives. Steganography is a technique of concealing information by skillfully embedding data within other media, such as images, audio, or text, without arousing suspicion. This is achieved through subtle modifications, like introducing alterations in pixel values or audio samples, rendering the hidden information nearly imperceptible to the human eye or ear. Steganography finds applications in various domains, including digital watermarking, covert communication, copyright protection, and data integrity verification, offering a range of tools and methods for enhancing information security and privacy protection. In [46], the authors employed the Blowfish algorithm and the Least Significant Bit (LSB) technique to conceal textual content within image files. First, encrypt the secret text using the Blowfish algorithm, and then conceal the encrypted secret text within an image using the Least Significant Bit (LSB) technique, thereby hiding confidential information in seemingly ordinary media or files to achieve confidentiality, secrecy, or Integrity verification purposes.

Shor’s algorithm [10], proposed in 1994, quickly dismantled the security of traditional algorithms such as RSA. This discovery drew attention to the threat that quantum computers pose to traditional encryption algorithms. The inaugural International Symposium on Post-Quantum Cryptography took place in 2006 at the University of Leuven. During this event, researchers and scholars from across the globe convened to explore cutting-edge advancements in quantum computing technology and cryptographic methods designed to withstand potential attacks from quantum computers. Both researchers and the attendees reached a consensus that, in the event of widespread availability of large-scale quantum computers, the significance of post-quantum cryptography will be paramount for shaping the future landscape of the Internet [47].

Subsequently, organizations such as NIST, the European Union, ETSI, IETF, and IEEE have played a significant role in promoting the study and standards specification of post-quantum cryptography. NIST has established a series of standard algorithms and specifications within the realm of cryptographic studies. A very large part of a series of algorithms and standards that are currently widely used internationally are specified by NIST, as shown in Table 4.

In 2012, the National Institute of Standards and Technology launched investigations into post-quantum cryptography. During April 2015, the International Conference on Practice and Theory of Public-Key Cryptography organized a Post-Quantum Workshop in Gaithersburg, Maryland. The aim was to address matters concerning post-quantum cryptography and explore its prospective standardization. Subsequently, NIST began working on specifying standards for post-quantum computing, and the timeline of the post-quantum cryptography process is shown in Figure 5.

In February 2016, NIST gave a talk at PQCrypto 2016, calling for anti-quantum encryption algorithms for new public-key cryptography standards and began a global call for PQC standards. In April of the same year, NIST presented at [48] and described the influence of quantum computing on traditional public-key encryption and symmetric encryption, provided an introduction to four different types of post-quantum encryption algorithms, and described quantum computer hardware developments, and concluded by pointing out directions for future research [49].

Ultimately, as of 30 November 2017, NIST received 82 draft algorithms. During the AsiaCrypt 2017 conference on 4 December 2017, NIST delivered a presentation titled “The Ship Has Said: The NIST Post Quantum Crypto ‘Competition’”. The presentation emphasized that with the advent of large-scale quantum computers, the necessity to transition from public-key cryptographic algorithms to post-quantum cryptographic algorithms would be crucial. Conversely, the urgency for addressing symmetric algorithms is less immediate, as adjustments to parameters can potentially provide solutions, as shown in Figure 6.

On 21 December 2017, NIST released 69 complete and suitable drafts from the first round after an initial screening, and the candidate algorithms were discussed, as shown in Table 5.

The first POC Standardization Conference with PQCrypto 2018 was held in Fort Lauderdale, Florida, on 11 April 2018, where the first round of candidates openly discussed and explained their proposed algorithms.

Following a year of comprehensive assessment, NIST unveiled its selection of 26 algorithms that advanced to the second round on 30 January 2019. This set encompasses 17 algorithms dedicated to public-key encryption and key establishment, alongside an additional 9 algorithms designed specifically for digital signature applications.

Subsequently, the NIST post-quantum cryptography standardization process entered the next phase. On 22 August 2019, the second PQC Standardization Conference, held in conjunction with PQCrypto 2019 in Santa Barbara, California, invited second-round candidates to submit brief updates on their algorithms and conducted discussions on various aspects of the candidate algorithms [50].

Quantum cryptography standardization entered its third phase after NIST announced its third round of candidates on 22 July 2020, which included seven candidate algorithms and eight alternates. The third PQC Standards Conference was held online on 7 June 2021 [41], where 15 algorithms, both candidate and alternate, were discussed. On 5 July 2022, NIST identified four standardized candidate algorithms: CRYSTALS-KYBER [51], CRYSTALS-DILITHIUM [52], FALCON [53] and SPHINCS+ [54]. Meanwhile, NIST announced the fourth round of candidates, including BIKE [55], Classic McEliece (a merger of Classic McEliece and NTS-KEM HQC) [56], HQC [57], and SIKE (with the SIKE team acknowledging that SIKE and SIDH are not secure) [58]. The fourth PQC Standards Conference was held online on 29 November 2022, where aspects of the candidate algorithms were discussed to inform standardization decisions, and submission teams for the selected algorithms, as well as the algorithm submission teams advancing to Round 4, were invited to update their algorithms.

In the future, post-quantum encryption will be combined with sensors to create more secure and efficient data transmission and protection systems. Quantum sensors, relying on the laws of quantum mechanics and utilizing effects such as quantum superposition, quantum entanglement, and quantum compression, possess extreme sensitivity and are able to monitor and capture minute physical and chemical changes in real time, such as the presence of viruses and cellular and molecular analysis. Post-quantum encryption algorithms will be the primary means of protecting sensor data against future quantum computer attacks, ensuring data confidentiality and integrity. This combination will be applied on a large scale in areas such as healthcare, environmental monitoring, and communications, providing a solid guarantee of data privacy while driving progress in scientific research and technological innovation.

## 3. CRYSTALS-KYBER Algorithm Construction Process

The CRYSTALS-KYBER algorithm was introduced by Peter Schwabe et al. in 2017 at the [51], in which it was published and identified as a standardized algorithm for post-quantum encryption in 2022. From a mathematical perspective, the CRYSTALS-KYBER algorithm is rooted in the concept of structured lattices. From another point of view, the CRYSTALS-KYBER algorithm is a Key Encapsulation Mechanism (KEM) algorithm based on the Module-LWE problem.

The Kyber algorithm provides encryption algorithms for IND-CPA (Chosen Plaintext Attack) safety and IND-CCA (Chosen Ciphertext Attack) safety. CCA security is a definition of safety in public-key cryptography, specifically proposed for public-key cryptography schemes. In public-key cryptography, users usually receive ciphertexts from multiple other communicating parties, which introduces a certain security risk. CCA safety aims to solve the security problem in this case. CPA safety is an important safety definition in modern cryptography for evaluating the reliability and safety of encryption algorithms. During encryption, CPA safety requires that the encryption algorithm be able to resist attempts by hostile approaches to attack the chosen plaintext. Although the two security requirements are different, they both focus on the attacker’s interaction with the encryption algorithm and aim to prevent an adversary attacker from obtaining useful information or cracking the encryption algorithm. The CCA variant of the Kyber algorithm is constructed upon the CPA version, employing a modification of the well-known Fujisaki–Okamoto transform. This adaptation is founded on the modular version [59] of the Ring-LWE LRP encryption algorithm [60] and introduces the integration of bit-discarding techniques [61,62].

The foundation of the Kyber algorithm rests on the Module-LWE problem [63,64]. Previous LWE-based cryptosystems have either used structured Ring-LWE problems (e.g., NewHope) or used standard LWE (e.g., Frodo [65]). The Module-LWE problem presents a variation of the LWE problem, wherein ring elements are substituted with module elements to enhance its adaptability [66]. Simultaneously expanding the problem space, Module LWE has efficiency and security advantages in specific situations, as well as utilizing different mathematical structures to provide enhanced encryption primitives.

Kyber belongs to active security. In [67], Bos et al. translated the migration of TLS to post-quantum security using passive secure KEM. Subsequently, e.g., NewHope [68] and Frodo [62] proposed more efficient and cautious examples of underlying passive security KEMs. Compared with active KEM, passive safety KEM can accept a higher probability of failure and does not require CCA transformation, enabling faster unpacking. However, experts define Kyber as IND-CCA2 secure KEM. Sometimes, active security is mandatory, such as in many applications such as public-key encryption or authenticated key exchange. Kyber’s CCA conversion is different from the secure scheme, as it can be noted that there is “all zero noise”, and errors can be immediately captured.

Kyber is built on the technique of number-theoretic transformations [69], and multiplication based on number-theoretic transformations has many advantages: (1) No additional memory is required (for example, like Karatsuba [70] or Toom [71] multiplication), and it can be quickly completed with very little code space. Therefore, the current common approach is to use lattice-based encryption parameters to support this multiplication algorithm [72]. And there are schemes that go a step further and include NTT as part of the scheme definition, such as [73,74]. In the case of public-key sampling, Kyber uses NTT as part of the scheme definition rather than the format of the ciphertext.

For the generation of the common uniform matrix A in Kyber, the “Against all authority” method in NewHope is used. The matrix is newly generated as part of each public key. This approach has two advantages: (1) it avoids the discussion of how uniformly random system parameters are actually generated, and (2) it prevents “one against all” attacks.

Kyber uses binomial noise. LWE-based cryptographic theories usually consider the use of Gaussian noise (circular Gaussian [75] or discrete Gaussian [76]) for LWE. Early implementations sampled noise from discrete Gaussian distributions, which were inefficient and susceptible to timing attacks. Therefore, Kyber relies on LWE instead of LWR as the fundamental issue.

Kyber allows for decryption failures. Kyber permits the choice of parameters that ensure not only minimal chances of unlocking failures but also complete elimination of such failures. Zero failure probability has some advantages, such as making CCA conversion and security proof easier to avoid attacks using unblocking failure. There are drawbacks to designing LWE-based encryption with zero failures, such as reducing the security or performance of attacks against underlying lattice problems.

### Kyber Build Specifics

In [51], Joppe Bos et al. from NXP commence by providing an abstract definition.

Kyber is actually an algorithm that uses asymmetric encryption to encapsulate keys and negotiate keys, with the key point on KEM, as shown in Figure 7.

In this context, “bi” represents the public key, “gi” represents the private key, “d” represents the ciphertext, and “i” represents the key.

The standard security concept based on public key cryptography, which is inseparable under the selection of ciphertext and selective plaintext attacks (IND-CCA and IND-CPA), in order to ensure the security of Kyber's encryption scheme, the advantages of op-ponent A are defined as:(3)$${\mathrm{Adv}}_{PKE}^{cca}(N)=\left|\mathrm{Pr}\left[\begin{array}{cc} u=u^{\prime }= & \begin{array}{l} (bi,gi)\leftarrow K\mathrm{ey}()\\ (y_{0},y_{1},g)\leftarrow A^{De(\cdot )}(bi);\\ u\leftarrow \{0,1\};d^{*}\leftarrow En(bi,y_{u});\\ u^{\prime }\leftarrow N^{De(\cdot )}(g,q^{*}) \end{array} \end{array}\right]-\frac{1}{2}\right|$$
where “*PKE*” represents the public-key encryption scheme, “*y*” represents the plaintext message to be encrypted, “*d*” represents the ciphertext generated after encryption, “*En*” represents the encryption function, “*g*” represents a uniform random variable, and “*De*” represents the decryption function.

According to the standard security concept of key encapsulation indistinguishability under selective ciphertext attacks, we define the advantage of opponent A as:(4)$${\mathrm{Adv}}_{KEM}^{cca}(N)=\left|\mathrm{Pr}\left[\begin{array}{cc} u=u^{\prime }= & \begin{array}{l} (bi,gi)\leftarrow K\mathrm{ey}()\\ u\leftarrow \left\{0,1\right\}\\ (d^{*},L_{0}^{*})\leftarrow Encap(bi);L_{1}^{*}\leftarrow L;\\ u^{\prime }\leftarrow N^{Decap(\cdot )}(bi,d^{*},L_{u}^{*}) \end{array} \end{array}\right]-\frac{1}{2}\right|$$
where “*KEM*” represents the encapsulated symmetric key, “*u*” represents the number of bits of the key, “*d*” represents the ciphertext generated after encryption, “*Encap*” represents the encryption function, “*Decap*” represents the decryption function, “*N*” stands for a uniform random matrix, and “*K*” stands for key space.

This measures A’s ability to distinguish between different information encryption or to retrieve encrypted data information.

Assuming that qt, qu, qv, k are all parameters represented by positive integers and n=256, then let O=0,1256 depict the range of messages, where each message can be regarded as a polynomial having coefficients 0,1 in the ring ℤY/(Yn+1). The Kyber_CPA=(Key,En,De) is described in Algorithms 1–3, and the algorithms are shown below.
**Algorithm 1.** 
Kyber_CPA_key():key generation
1. $\mu ,\delta :\leftarrow {\left\{0,1\right\}}^{256}$
2. $N\sim R_{t}^{i\times i}:=S(b)$
3. $\left(g,j)\sim \beta_{\eta }^{i}\times \beta_{\eta }^{i}:=S(\delta \right)$
4. $f:=Comp_{t}(Ng+j,q_{t})$
5: return (bi:=(f,b),gi:=g)


**Algorithm 2.** 
Kyber_CPA_En(bi=(f,μ),y∈O):encryption
1. γ←0,1256
2. f:=Decompt(f,df)
3. N~Rti×i:=S(μ)
4. (γ,e1,e2)~βηi×βηi×βη:=S(γ)5. α:=Compt(NTγ+e1,qα)
6. β:=Compt(fTγ+e2+t2)⋅y,tβ)7: return d:=(α,β)


**Algorithm 3.** 
Kyber_CPA_De(gi=g,d=(α,β)):decryption
1. $\alpha :=Decomp_{t}(\alpha ,q_{\alpha })$
2. $\beta :=Decomp_{t}(\beta ,q_{\alpha })$
3: return d:=(α,β)


The author proves it and demonstrates the validity and safety of the encryption method. In conclusion, the encryption scheme outlined earlier achieves IND-CPA security, relying on the principles of Module LWE.

Adopted by [77], an altered form of the Fujisaki–Okamoto transformation in [58] is applied to the Kyber.CPA encryption scheme to obtain a CCA-secure KEM.

The algorithm can be described by Algorithms 1, 4 and 5, with Algorithms 4 and 5 as shown below.
**Algorithm 4.** 
Kyber_Encap(bi=(b,f))
1: $y\leftarrow {\left\{0,1\right\}}^{256}$
2: $\left(\hat{L},\gamma^{\prime },q^{\prime }):=G(gi,y^{\prime }\right)$
3: $\left(\alpha ,\beta ):=Kyber\_CPA\_En((\mu ,f),y^{\prime };\gamma^{\prime }\right)$
4: d:=(α,β,q)
5: $L:=H(\hat{L},d)$
6: return (d,L)


**Algorithm 5.** 
Kyber_Decap(gi=(g,z,α,f),c=(α,β,q))
1. $y^{\prime }:=Kyber\_CPA\_De(g,(\alpha ,\beta ))$
2. $\left({\hat{L}}^{\prime },\gamma^{\prime },q^{\prime }):=G(gi,y^{\prime }\right)$
3. $\left(\alpha^{\prime },\beta^{\prime }):=Kyber\_CPA\_En((\mu ,f),y^{\prime };\gamma^{\prime }\right)$
4: if $\left(\alpha^{\prime },\beta^{\prime },q^{\prime })=(\alpha ,\beta ,q\right)$ then5: return $L:=H({\hat{L}}^{\prime },d)$
6: else7: return L:=H(z,d)
8: end if

In [58,59], in which it is shown that if Kyber.CPA is secure, then Kyber is CCA secure under the quantum stochastic predicator model.

Subsequently, the parameters and security are analyzed, and in order to achieve quantum (and classical) security after 128 bits and to cope with future improvements in cryptanalysis, the set of Kyber parameters is proposed after considering only the parameters related to the basic lattice problem, as shown in Table 6.

Determine functions H and G based on the previous abstract definitions, one for accepting the public seed as input and generating the uniform matrix and the other for accepting the cipher seed r as input and generating the sampled noise polynomial as output. In passively secure KEMs such as BCNS [67], NewHope [68], and Frodo [77], the choice of the noise polynomial sampling method is a localized strategy; implementations on different platforms can choose the best PRNG on their respective platforms. The Kyber algorithm adopts the CCA transform, so all hash functions are instantiated using the FIPS 202 standard scalable output function SHAKE-128. The performance results of Kyber were compared with other literature, and the comparative outcomes are displayed in Table A1.

The above is the original version of Kyber, and the algorithm was tweaked twice within the course of the NIST PQC standardization process’s second and third rounds.

The first change to the Kyber core design is as follows:Increase the noise parameter of Kyber512.

In the second round of submission by Kyber, faced with the relatively conservative decryption error of Kyber512 and the need to increase the difficulty of the parameter set Core SVP, it was decided to increase the binomial error distribution of Kyber512. To avoid increasing noise, a method similar to the LWR assumption is adopted, which relies on rounding noise to increase error.

2.Reduce the ciphertext compression rate of Kyber512.

There is a positive correlation between noise and the likelihood of unsuccessful decryption attempts. Increasing noise increases the likelihood of unsuccessful decryption attempts, reducing the quantity of bits located in the “second” ciphertext element. The size of the encrypted message has been increased to 768 bytes, and the decryption error rate is 2.

3.Common matrix A adopts more effective uniform sampling.

The use of rejection sampling on 12-bit integers is used to replace the previous use of rejection sampling on 2-byte integers to sample uniformly random integers of mode 3329 in the district.

The next changes to the specifications and supporting documentation are as follows:Updated specifications to match Round 3 parameters.Updated performance data.The performance analysis section includes data on ARM Coretex-M4 [78], and the submitted software package includes the corresponding software. Finally, a more detailed analysis was conducted on the latest technologies to solve the core Gaussian programming problem, and attacks that failed decryption were discussed. Significant updates were made in terms of security analysis.

After the National Institute of Standards and Technology and cryptographers around the world evaluated and validated various encryption methods and concluded that the Kyber algorithm has the capacity to endure impending quantum computer-based attacks, the Kyber algorithm was selected by NIST as a standardized algorithm. The Kyber algorithm, as a part of the Post-Quantum Cryptography Standard, can provide a more secure and reliable encryption method to ensure that digital information can still be effectively protected under the threat of future quantum computers, maintaining the trust and confidentiality [79] of digital systems such as online banking and email that are used on a daily basis, and providing society with an effective tool to deal with future threats to enhance the security and stability of digital information [80].

## 4. Implementation

The implementation of cryptographic algorithms can be divided into two main forms: software implementation and hardware implementation. Software implementation involves converting the design of a cryptographic algorithm into the form of a software program, usually implemented through a programming language. In software implementation, the running of the algorithm is done on a general-purpose computing device, such as a personal computer, server, or mobile device. Advantages of this approach include relative ease of development and deployment, as it can run on a wide range of computing platforms but may suffer from computational resource constraints, resulting in slower speeds in some cases. Hardware implementation involves embedding cryptographic algorithms into specially designed hardware devices to achieve a more efficient and faster encryption and decryption process. These specialized hardware devices are often called cryptographic coprocessors or cryptographic accelerators. The advantage of hardware implementations is that they can provide higher performance and lower latency because they are specifically optimized for cryptographic operations. This is important for applications that require a high degree of security and speed, such as network equipment, cloud computing servers, and embedded systems. The summary diagram of software implementation and hardware implementation is shown in Table 7.

### 4.1. Software Implementation

In 2019, Leon Botors et al. [81] presented an optimized software implementation of the Kyber algorithm designed for a 32-bit RISC processor core ARM Cortex-M4 microcontroller. At the core is a novel optimization technique for Kyber’s internal Number Theoretic Transform (NTT) that efficiently exploits the computational power provided by the target architecture’s “vector” DSP instructions. The solution first optimizes the speed of NTT, which greatly improves the implementation speed of Kyber, implementing Kyber with less stack space and computational overhead. Compared with the first and second versions of Kyber, it was concluded that compared to other candidate algorithm implementations on ARM Cortex-M4 (as shown in Table A2), this approach to implementing Kyber minimizes memory consumption and results in the fewest total cycles required for tasks such as key generation, encapsulation, and unpacking. A performance disparity exists between the most rapid implementation of Saber, as documented in [87], and the implementation optimized for stack usage outlined in [88]. Finally, compared to the Kyber algorithm software implementation, it is 18% faster.

Joppe W. Bos and colleagues, as presented in [82], introduced the initial fully masked implementation of Kyber. In order to achieve complete Kyber first-order and high-order masks, a new mask algorithm of two modules is proposed. New masking algorithms are proposed for two modules: (1) Masked single-bit compression: previous solutions were either limited to first-order masks or compressed using power-of-2 modes. The author proposes a new method of binary search. (2) Comparison of masked decompression: Kyber uses ciphertext compression techniques. It can effectively mask power-of-two modes but imposes a non-negligible overhead for prime modes. The writer introduces a novel approach for comparing uncompressed masked polynomials with compressed public polynomials, eliminating the need for explicit mask compression of ciphertext by combining known techniques with the two new methods to realize a masking scheme for Kyber complete decapsulation. Numerous experiments have shown that this method is effective. By comparison, on the Cortex-M0+ platform, compared with the original shielded Kyber version, the speed is reduced by 2.2 times; on the Cortex-M4F platform, the first-order unenhanced implementation of the optimized polynomial algorithm assembly routine is used, which is the same as the optimized pqm4. Compared with the previous version of Kyber, the speed is reduced by 3.5 times.

### 4.2. Hardware Implementation

In [85], NANNIPIERI et al. studied the potential hardware acceleration of the Kyber algorithm. They proposed the first post-quantum ISA encryption extension to a 64-bit CVA6 RISC-V processor and reduced crystal kit execution time by introducing new hardware functions that map directly to assembly instructions. Through testing, it was found that an acceleration of 20% to 65% was achieved.

In 2021, Xing et al. introduced, in [86], a manually designed Kyber hardware implementation, integrating the CCA secure key exchange mechanism CRYSTALS-KYBER into a compact hardware programmable gate array FPGA. The protocol is optimized from a hardware and algorithmic point of view to implement it. Methods to minimize the memory footprint are discussed. After that, the document presents performance outcomes on specific FPGA devices and offers a comparative analysis of relevant prior research. The researchers in this study introduced a hardware implementation of CRYSTALS-KYBER that is entirely hardware-based. During the implementation process of this solution, soft cores implemented with reconfigurable logic, such as ARM Cortex series and popular RISC-V processors, were not used. They utilized limited resources to achieve good performance. Considering that the NTTs in Kyber are slightly different, one of the 256 NTTs can be seen as two separate 128 NTTs; one is an odd index, and the other is an even index. Two sets of butterfly-shaped devices are used to handle the even and odd parts, respectively. Through experiments, the achievable performance in independent hardware design was revealed. It has better performance than solutions implemented using software and hardware co-design or HLS methods.

## 5. Discussion and Evolution

The advancement of post-quantum cryptographic algorithms is currently making encouraging strides. The National Institute of Standards and Technology has unveiled the initial four standardized post-quantum encryption algorithms as part of its Post-Quantum Cryptography Standardization Project. This achievement is seen as an important milestone, marking the initial success of efforts to protect the privacy of digital systems in the context of future capabilities to break current encryption algorithms. The four encryption algorithms identified for standardization are expected to be finalized within two years. These algorithms will provide important support for future encryption in the post-quantum era, ensuring that digital communications and information security will not be threatened by quantum computers in the future [89].

In recent years, quantum computers have been continuously developing. Abroad, IBM released a roadmap for quantum computing technology in 2020, aiming to exceed 100 qubits by 2021 and 1000 qubits by 2023, ultimately leading IBM to quantum computing devices at the level of millions of qubits or higher. In 2020, Google released a plan to implement 1 million physical quantum bit processors by 2029. Google stated that they have developed plans to gradually expand quantum processors with milestones of 102, 103, 104, 105, and 106 quantum bits. A computer with one million qubits will consist of 100 modules, each containing 100 × 100 quantum bits. In China, the academic leader of Huawei’s quantum computing research is Professor Wenkang Weng, whose main research direction is specialized quantum computing algorithms and software for the NISQ era. Huawei released a new public beta version of the HiQ quantum computing cloud platform in 2021, achieving phased research results in quantum simulators and programming frameworks [90]. The Alibaba team questioned the superiority of Google Quantum and developed a simulation algorithm based on tensor networks. The algorithm was tested on Alibaba Cloud and compared to supercomputer Summit’s cluster approach; it showed that it could solve the 10,000-year version of the random quantum line sampling problem used in Google testing in just 20 days (53 qubits, 20 cycles) [87].

It is not difficult to see that quantum computers are developing rapidly and will become popular in the future.

### 5.1. Opportunities and Challenges

In the post-quantum encryption algorithm field, there are many promising opportunities and challenging tasks. On one hand, there are opportunities presented by the emergence of post-quantum encryption algorithms [91].

The development of post-quantum encryption algorithms effectively guards against quantum computer threats, providing a robust foundation for future information security. This is crucial for safeguarding information in various industries, including finance, telecommunications, healthcare, and more.Research in post-quantum encryption has also driven the establishment of encryption standards and international collaboration, with the potential to provide a consistent foundation for ensuring global information security.

On the other hand, with the emergence of the concept of post-quantum encryption algorithms, this field presents numerous challenges [92].

Developing sufficiently strong and efficient post-quantum encryption algorithms is a massive undertaking that requires in-depth mathematical and computer science research. Large-scale deployment and standardization will involve complex coordination and cooperation to ensure widespread adoption of new technologies.To maintain security, it is essential to continuously address potential new threats and attacks, which requires staying vigilant and updating encryption systems promptly.

In summary, post-quantum encryption algorithms represent a revolutionary advancement in the field of information security, providing critical protective mechanisms for the future [93]. However, there are complex challenges in terms of research, development, deployment, and maintenance that require interdisciplinary collaboration and continuous innovation to address [94].

### 5.2. Evolution

Currently, the timeline for the construction of large-scale quantum computers remains uncertain. Some scientists predict the future based on the current situation, believing that large-scale quantum computers will emerge in 20 to 30 years and that the powerful computing power of large-scale quantum computers will crack most existing public-key encryption algorithms. Therefore, in order to maintain information security, security systems that use new encryption algorithms to resist large-scale quantum computers should start preparing as soon as possible [88].

In the coming years, as post-quantum encryption continues to evolve, the anticipated trends are as follows:Adoption and standardization of new encryption algorithms: With NIST establishing standardized post-quantum encryption algorithms, new opportunities are provided for data security and communications in various fields [95]. The anticipated future trends in post-quantum encryption algorithms are illustrated in Figure 8.Evolution of crypto: Post-quantum encryption is not just about responding to the threat of quantum computers but will drive the evolution of the entire field of encryption. This could include stronger authentication, more sophisticated key management, and more efficient encryption protocols. These evolutions will help improve the security of communications and data and adapt to evolving threats [96].Practicality and performance considerations: Post-quantum encryption algorithms need to be secure while also considering their practicality and performance. These algorithms need to be able to be used in real communications without significantly affecting the speed and efficiency of communications. Therefore, future research and development will focus not only on security but also on the practicality and deployability of the algorithms [97].

Today, the Internet of Things has profoundly changed our way of life. It allows us to remotely control home equipment, such as temperature, lighting, etc., to realize smart homes and improve the quality of life. At the same time, smart monitoring equipment can also monitor our physical health in real time to facilitate disease prevention. As one of the key components of the Internet of Things, sensors are also crucial to ensuring the security of sensor data [98].

A sensor is a device used to perceive and detect various types of information in the environment, such as physical, chemical, and biological data. It converts this information into electrical signals or digital data for processing, analysis, and control by computers or other systems [99]. Sensors have a wide range of applications, including but not limited to monitoring physical phenomena like temperature, humidity, pressure, light, sound, and motion, as well as detecting chemical components and biological indicators. They play a crucial role in fields such as industrial automation, medical diagnostics, environmental monitoring, smartphones, automobiles, robots, and more. Sensors come in various types, including temperature sensors, pressure sensors, optical sensors, accelerometers, gyroscopes, and biometric sensors, each with specific principles of operation and application areas. The continuous development and innovation in sensor technology drive advances in science and engineering, enabling us to better understand and utilize the world around us. All data play an important role in various fields. Firstly, sensor data is used for decision making and controlling various systems, including industrial automation, traffic systems, and medical devices. Tampering with or malicious interference with these data can lead to serious accidents and losses. Secondly, sensor data are widely used in scientific research for monitoring environmental changes, weather forecasting, and geological exploration, among other fields [100]. The accuracy and integrity of data are essential for the reliability of research. Additionally, sensor data are utilized for monitoring the vital signs of medical patients, and any data tampering could pose a serious threat to their health. Finally, with the proliferation of the Internet of Things (IoT), sensor data will involve more personal privacy information, including home security and health data. Therefore, ensuring the confidentiality, integrity, and availability of sensor data is crucial for maintaining the normal operation of systems, the credibility of research, and the protection of personal privacy [101].

In 2015, NIST initiated the Lightweight Cryptography (LWC) project with the aim of researching and promoting lightweight cryptographic algorithms for resource-constrained devices, such as sensors and IoT devices. After a series of selection processes, ASCON was eventually established as the standardized algorithm for lightweight cryptography. In 2016, NIST launched the Post-Quantum Cryptography (PQC) project, which aims to identify new encryption algorithms capable of withstanding attacks from quantum computers.

The algorithms within the LWC project possess characteristics of lightweight, efficiency, and security, making them well-suited to provide security for resource-constrained devices. Algorithms from the PQC project are capable of withstanding attacks from quantum computers while maintaining excellent efficiency. As quantum computing continues to advance, adapting standardized post-quantum encryption algorithms for resource-constrained devices is a crucial research direction for the future [102]. This can be approached from three perspectives:Sensors typically transmit their collected data to other devices or storage servers. Consider employing post-quantum encryption algorithms to encrypt sensor data, ensuring data security during transmission.Sensors often require identity verification to ensure legitimacy. Post-quantum encryption algorithms can be used for sensor identity verification, ensuring data integrity and authenticity.Sensors may occasionally need firmware updates and configuration changes. Post-quantum encryption algorithms can be utilized to verify the integrity of sensor firmware and configuration files, ensuring the security of the sensors.

In the future, applying post-quantum encryption algorithms to sensors will bring higher security to the sensors, prevent the data generated by the sensors from being cracked and tampered with, and better protect sensitive data.

## 6. Summary and Future Work

Today, there is a growing demand for data security. Post-quantum encryption algorithms are substantially more secure compared to traditional encryption algorithms, which can better safeguard the security of sensor data.

Post-quantum encryption is in an active stage of development. These next-generation cryptographic techniques can be nested in various parts of the network to provide an additional layer of safety for the whole network to withstand the information security challenges in the quantum computing era [5]. Meanwhile, modern cryptography systems based on mathematical algorithms are quite mature and widely used, but password-cracking techniques are still evolving. Post-quantum cryptography, deemed a novel generation of cryptographic algorithms, is designed to withstand assaults from quantum computers. It is progressively supplanting conventional public-key cryptographic methods like RSA, Diffie–Hellman, and elliptic curves. Anticipated to gain prominence within the next 5–10 years, it is poised to emerge as a prevailing trend. The U.S. National Institute of Standards and Technology is at the forefront of shaping a new era of cryptographic standards known as post-quantum cryptography standards. This pivotal role has a significant impact on propelling advancements within this domain. The European Union’s Network Security Agency is also exploring ways to integrate post-quantum cryptosystems into existing protocols, as well as designing new protocols to cope with the demands of post-quantum systems. The field of post-quantum cryptography faces many challenges in terms of technology and standardization, but it also has great potential for growth. Post-quantum encryption is driving a revolution in the field of cryptography to guarantee information safety in the quantum age.

The advent of post-quantum encryption algorithms can better secure sensor data. This leads to better privacy protection, prevention of data leakage, ensuring data integrity and reliability, etc., as well as better use of sensor data for decision making and innovation.

At the end of 2022, after several rounds of selection, NIST selected Kyber as the standard for post-quantum encryption algorithms because it believed that the Kyber algorithm not only possessed better security but also that Kyber could achieve very good performance on hardware and software implementations on multiple platforms and could be well embedded into most existing Internet protocol/cryptographic algorithm applications.

Although quantum computers that can actually break these algorithms are not yet commonplace, post-quantum cryptography has become inevitable for future security and is already being progressively implemented. To foster the advancement of post-quantum encryption, NIST is committed to developing post-quantum encryption standards. After the new algorithms and standards are ready, the adoption and migration to post-quantum cryptography will require consideration of many factors, including updates to protocols, programs, and infrastructure. NIST will also be holding discussions on plans to migrate to post-quantum cryptography and proposes next steps to assist in the migration. While there are many practical challenges that may be faced in introducing post-quantum cryptographic schemes, such as algorithmic performance and speed of deployment, these efforts are aimed at better protecting people’s communications and data and ensuring the security and privacy of future network communications [103]. While quantum computers may pose a threat, they also provide opportunities to reflect, discover, and build better security, opening a new chapter in the development of post-quantum encryption.

## Figures and Tables

**Figure 1 sensors-23-08744-f001:**
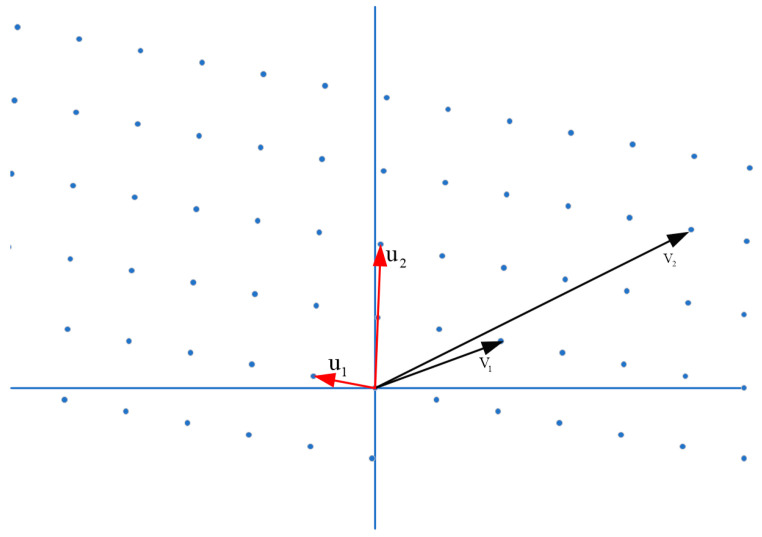
Two-dimensional grids and two different sets of grids.

**Figure 2 sensors-23-08744-f002:**
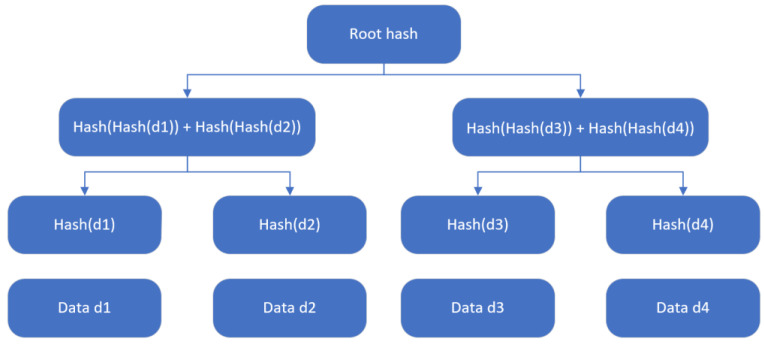
Merkle tree.

**Figure 3 sensors-23-08744-f003:**
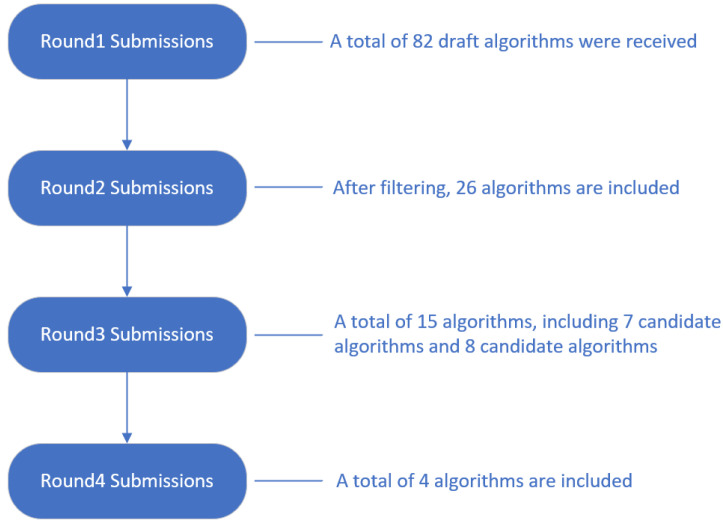
NIST four rounds of screening.

**Figure 4 sensors-23-08744-f004:**
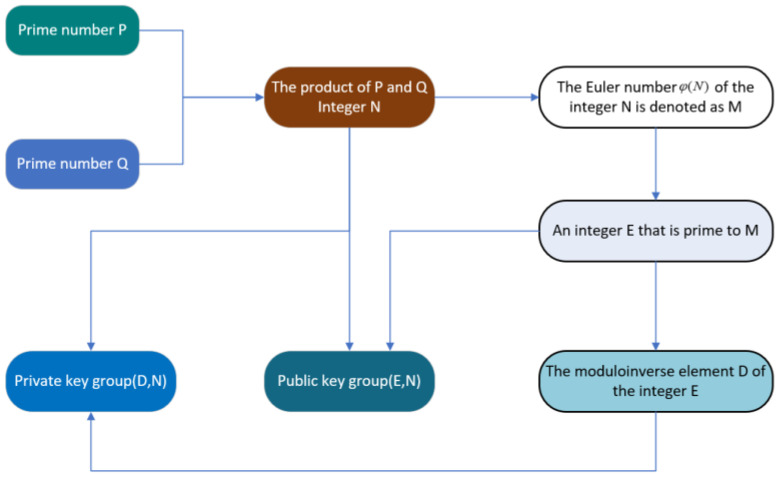
RSA algorithm key generation process.

**Figure 5 sensors-23-08744-f005:**
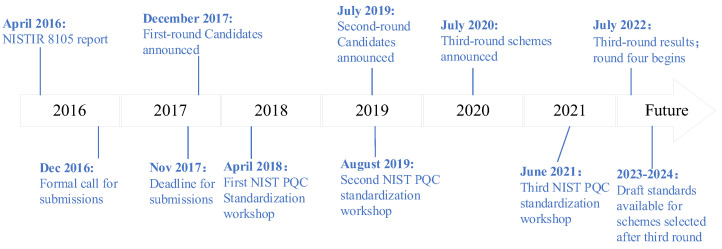
Timeline of the post-NIST quantum cryptography process.

**Figure 6 sensors-23-08744-f006:**
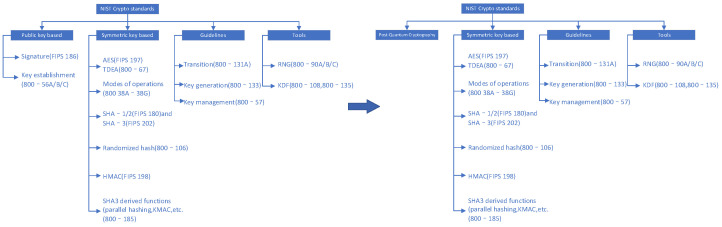
Future direction of NIST encryption standards.

**Figure 7 sensors-23-08744-f007:**
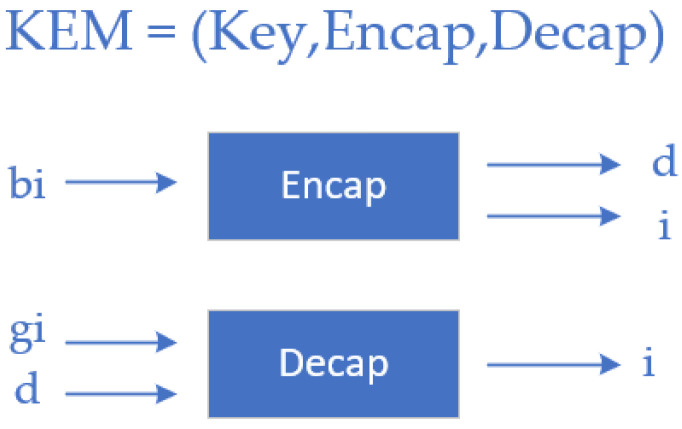
Key encapsulation mechanism.

**Figure 8 sensors-23-08744-f008:**
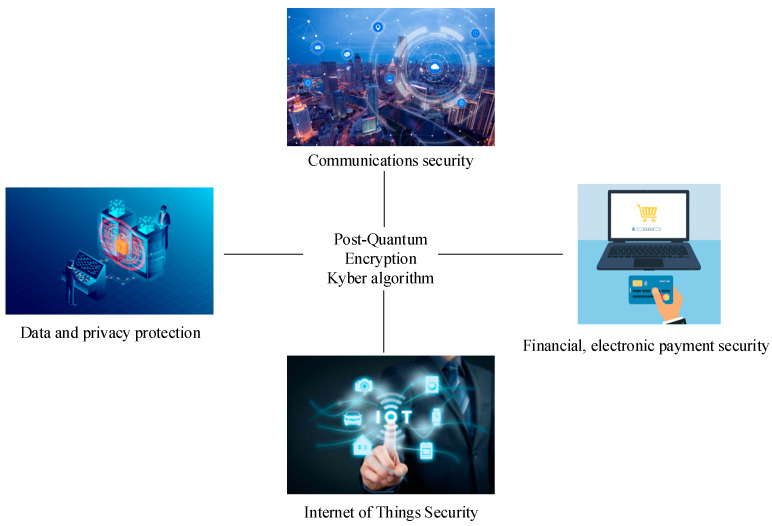
Background of post-quantum encryption Kyber algorithm applications.

**Table 2 sensors-23-08744-t002:** Classification of error correction codes under different standards.

	Specific Classification
Based on the relationship between the information element and the check factor	(1) Linear code (2) Nonlinear code
The type of correction based on the error code element	(1) Correct random error codes (2) Correct error codes (3) Correct synchronization error codes (4) It can correct both random errors and sudden error codes
Based on the processing of information elements	(1) Convolutional code (2) Block code
Based on the relationship between codewords	(1) Cyclic code (2) Acyclic code

**Table 3 sensors-23-08744-t003:** Advantages and disadvantages of 4 post-quantum encryptions.

Name	Advantages	Drawbacks
Lattice-based Cryptography	Based on lattice theory, security builds on number-theoretic puzzles; extensive research support and proven algorithms.	The computational intricacy of encryption and decryption is relatively high; the key and message lengths are long.
Code-based Cryptography	There has been a long history of research based on the difficult problem of error-correcting codes; relatively short key lengths.	Higher computational complexity for encryption and decryption; longer key and signature lengths.
Multivariate-based Cryptography	Difficult problems based on systems of polynomial equations that are mathematically rigorously defined and understood. This provides a solid theoretical foundation for security.	Encryption and decryption performance is relatively poor, requiring more computational resources and time. This may have an impact on the practical feasibility of certain application scenarios.
Hash-based Cryptography	Relatively mature schemes such as one-time signature schemes based on cryptographic hash functions (Merkle–Damgard constructs) are already available; shorter key and signature lengths.	The computational complexity of public-key exchange is high; it requires a long signature verification time.

**Table 4 sensors-23-08744-t004:** Examples of standard algorithms and specifications established by NIST.

Algorithm	Instance
symmetric encryption algorithm	AES, DES, 3DES
digital signature algorithm	RSA, DSA, ECDSA
key exchange algorithm	Diffie–Hellman, ECDH, MQV
hash function	SHA-1, SHA-2 Series, SHA-3 Series
message authentication code	HMAC, KMAC, CMAC
Random number generation	Hash_DRBG, CTR_DRBG, Dual_EC_DRBG, HMAC_DRBG

**Table 5 sensors-23-08744-t005:** Ryo Fujita statistics from NIST post-quantum cryptographic algorithm standard solicitation round 1.

Category	Name	Overall
Lattice-based	CRYSTALS-DILITHIUM, DRS, FALCON, PqNTRUSign, qTESLA, Compact LWE, CRYSTALS-KYBER, Ding Key Exchange, EMBLEM and REMBLEM, FrodoKEM, HILA5, KCL, KINDI, LAC, LIMA, Lizard., LOTUS, NewHope, NTRU-HRSS-KEM, NTRU Prime, Odd Manhattan, Round2, SABER, Three Bears, Titanium	26
Code-based	PqsigRM, RaCoSS, BIG QUAKE, BIKE, Classic McEliece, DAGS, HQC, LAKE, LEDAkem, LEDApkc, Lepton, LOCKER, McNie, NTS-KEM, Quroboros-R, QC-MDPC KEM, RLCE-KEM, RQC	18
Multi-variate	DualModeMS, GeMSS, Gui, HiMQ-3, LUOV, MQDSS, Rainbow, CFPKM, DME	9
Symmetric/ Hash-based	Gravity-SPHINCS, Picnic, SPHINCS+	3
Other	Post-quantum RSA-Signature, WainutDSA, Giophantus, Guess Again, Mersenne-756839, Post-quantum, RSA-Encryption, Ramstake SIKE	8
Total		64

**Table 6 sensors-23-08744-t006:** Kyber parameter set.

	*n*	t	i	η	δ	$\left(q_{\alpha },q_{\beta },q_{f}\right)$	bt−sed
Kyber	256	7681	3	4	$2^{-142}$	(11,3,11)	161

**Table 7 sensors-23-08744-t007:** Summary of software implementation and hardware implementation.

Work	Measure	Effect
**Software Implementation**		
[81]	An improved optimization technology for NTT in Kyber is proposed.	Through experimental comparison, the improved software is 18% faster than the original author’s software. NTT in the improved software is more than twice as fast as the original software.
[82]	Kyber’s first complete masking was achieved by introducing two new technologies: Masked One-Bit Compression and Masked Decompressed Comparison.	The first masking scheme for complete Kyber decapsulation is proposed. Through experimental comparison, the improved software’s resistance to attacks has been improved.
[83]	Improve a technique to speed up OR operations at increased post-processing cost by replacing arithmetic multiplication with Galois field multiplication.	Through experimental comparison, the improved method is better than the original method. Implementation-specific improvements increase direct comparison implementation speed by 33%.
[84]	Kyber is optimized by combining different existing methods. Performance data were independently verified, and first-order resistance was confirmed using fixed and randomized TVLA methods.	A first-order shielded Kyber specifically for ARM CortexM4 is proposed and the first-order resistance is actually verified using ChipWhisperer Lite. It has been experimentally concluded that first-order masking is not sufficient to achieve practical side-channel immunity.
**Hardware implementation**		
[85]	Embeds post-quantum ISA cryptographic extensions into the 64-bit CVA6 RISC-V processor and introduces new hardware features that map directly to assembly instructions.	Through experiments, we demonstrate that PQC algorithms can be significantly accelerated by leveraging the flexibility of RISC-V processors and integrating dedicated accelerators directly into the core pipeline.
[86]	Kyber’s independent hardware design is proposed through the FPGA platform. Through careful scheduling of sampling and Number Theoretic Transform (NTT)-related computations, good performance can be achieved with limited hardware resources.	Through careful scheduling and noise sampling of NTT-related procedures, the design achieves good performance and can be installed in the smallest devices in the Xilinx Artix-7 series FPGAs. Through experiments, the computational efficiency of the scheme based on structured lattice is verified.

## Data Availability

No new data were created.

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
