# Peer review of "Post-Quantum Security: Opportunities and Challenges"

_sensors, 2023, doi:10.3390/s23218744_

Round 1
Reviewer 1 Report
This article can help readers gain a better understanding of post-quantum encryption techniques and their potential future significance. and has a thorough understanding of the Kyber construction principle and the direction that post-quantum encryption will take in the future. Although this work could provide valuable and interesting information for readers in this journal, some following concerns must be addressed:
1- “Johannes A. With the advancement of technology, sensors are finding increasingly widespread applications in daily life.” This sentence needs to be cited by its reference.
2- Table 1. must be at the bottom of the page because the bottom of the page contains a reference number that is less than what is found in Table 1.
3- In the introduction section, the fourth paragraph has two sentences and is not considered a paragraph, I suggest merging it with the previous paragraph.
4- In the introduction section, the last sentence of the sixth paragraph needs to be cited by its reference.
5- In the introduction section, the second paragraph of the fourth point has two sentences and is not considered a paragraph, I suggest merging it with the previous paragraph.
6- “After four rounds of screening, as shown in Figure 3.” To which paragraph does this sentence belong?
7- “Because Kyber has only recently been established.....” This paragraph explains the motivation of this paper, but it is one sentence and lacks information that explains the main goal of this paper.
8- “Today, we are in the digital age, where data is…” All information in this paragraph is mentioned in the first paragraph. I suggest removing this paragraph or changing.
9- Add a paragraph at the end of the introduction has explains What the article covers in the following sections.
10- In the background section, the first paragraph has two sentences and is short, I suggest merging it with the next paragraph.
11- “The most basic cryptographic classification is the division of message encryption into Symmetric Cryptography and Public-Key Cryptography Asymmetric Cryptography.” This is not a paragraph; I suggest merging it with the previous paragraph.
12- Merge the two paragraphs of 'Symmetric encryption' in one paragraph.
13- “The RSA algorithm is described as follows:” This is not a paragraph; I suggest merging it with the previous paragraph and the next sentence makes like a list of numbers or bullets.
14- “if e and are known, d is easy to calculate;” I think you missed something in this sentence, I suggest, correcting it.
15- “The encryption algorithm is” and “The decryption algorithm is” These sentences are not complete, please complete them.
16- Figure 5., Figure 6 and Figure 8., please increase the resolution of them.
17- Can you make a one-paragraph from "Subsequently, on December 21, 2017 ..." to "... the candidate algorithms were discussed."?
18- There is no caption for the first column of Table 5. Please, add it.
19- “Quantum cryptography standardization entered its third phase after NIST announced its third round of candidates on July 22, 2020, which included seven candidate algorithms and eight alternates.” This sentence is not considered a paragraph, I suggest merging it with the next paragraph.
20- “, where the second PQC Standardization Conference with PQCrypto 2019 was held on August 22, 2019 in Santa Barbara, California, where second-round candidates were invited to submit short updates to the algorithms and aspects of the candidate algorithms were discussed [46].” I think you missed something in this sentence, I suggest correcting it.
21- “[51], and announced the fourth round of candidates BIKE [52], Classic McEliece (merger of Classic McEliece and NTS-KEM HQC) [53], HQC [54], SIKE (the SIKE team recognizes that SIKE and SIDH are not secure) [55] .” I think you missed something in this sentence, I suggest correcting it.
22- “The fourth PQC Standards Conference was held online on November 29, 2022, where aspects of the candidate algorithms were discussed to inform standardization decisions, and submission teams for the selected algorithms, as well as the algorithm submission teams advancing to Round 4, were invited to update their algorithms.” This sentence is not considered a paragraph, I suggest merging it with the previous paragraph.
23- What do CCA and CPA mean? please mention the full term of the abbreviations.
24- “The CCA variant of the Kyber algorithm is …. ” This paragraph has two sentences and is not considered a paragraph, I suggest merging it with the previous paragraph.
25- On page 12, please explain the parameter in 1 and 2.
26- “The is described in terms of Algorithms 1, 4 and 5. Algorithms 4 and 5 are shown below.” I think you missed something in this sentence, I suggest correcting it.
27- In the Implementation section, please check the first and second paragraphs.
28- Are the software implementations in [81],[82], [83], and [84] the same? If not, please mention the main difference between them.
29- Are the software implementations in [85], [86] and [87] the same? If not, please mention the main difference between them.
30- Can you merge with the previous paragraph from "The application context of the post-quantum ..." to "... t anticipated trends are as follows:"?
31- Opportunities and Challenges of Post-quantum Security must appear as a list of numbers clearly in a separate section.
32- I did not see any mention of modern algorithms used in steganography as the Blowfish algorithm so I suggest reading about it in (http://doi.org/10.11591/ijeecs.v29.i1.pp339-347) and adding a paragraph about it.
33- Some of the references used are old. Can it be replaced with modern references?
Author Response
Dear Reviewer:
Thank you for your letter and for the reviewers’ comments concerning our manuscript entitled “Post-quantum Security: Opportunities and Challenges”. Those comments are all valuable and very helpful for revising and improving our paper, as well as the important guiding significance to our researches. We have studied comments carefully and have made correction which we hope meet with approval.
Revised portion are marked in yellow in the paper. The main corrections in the paper and the responds to the reviewer’s comments are as flowing:
- In the light of the content of recommendation No. 1, a re-reading of the article reveals an error in the text of "Johannes A", which has been deleted.
- Place the table at the bottom of the page as per suggestion #2.
- In accordance with the fourth recommendation, a reference has been added after the last sentence of the sixth paragraph.
- Based on the seventh recommendation, the article was restructured to reinterpret the information about the main goal of the paper.
- In accordance with recommendation 9, a paragraph has been added at the end of the introduction explaining the content of the following parts of the paper.
- Corrected the content of the RSA process in accordance with recommendation 14.
- Complete the sentence "The encryption algorithm is" in accordance with recommendation 15.
- The resolution of the images in the text has been increased in accordance with recommendation 16.
- The sentence has been modified in accordance with recommendation 17.
- In accordance with recommendation 18, the heading of the first column of table 5 has been added.
- The error in the content of ", where the second PQC Standardization Conference with PQCrypto 2019 was held on August 22, 2019 in Santa Barbara, California, where second-round candidates were invited to submit short updates to the algorithms and aspects of the candidate algorithms were discussed [46]." has been corrected in accordance with recommendation 20.
- The error in the content of " [51], and announced the fourth round of candidates BIKE [52], Classic McEliece (merger of Classic McEliece and NTS-KEM HQC) [53], HQC [54], SIKE (the SIKE team recognizes that SIKE and SIDH are not secure) [55]." has been corrected in accordance with recommendation 21.
- Labelling the specific names of CCA and CPA in the article in accordance with recommendation 23.
- The parameters of the two formulas are explained in accordance with recommendation 25.
- The error in the content of " The is described in terms of Algorithms 1, 4 and 5. Algorithms 4 and 5 are shown below." has been corrected in accordance with recommendation 26.
- In accordance with recommendation 27, the first two paragraphs of the operative part have been checked and the errors corrected.
- In accordance with recommendations 28 and 29, the implementation section has been reorganised into two parts, "software implementation" and "hardware implementation", and is briefly described.
- In accordance with recommendation 31, a section on "Opportunities and Challenges" has been added to Part IV.
- In accordance with recommendation 32, steganography and the Blowfish algorithm were added to the presentation of traditional encryption algorithms.
- In accordance with recommendation 33, many of the older references were replaced after reviewing the relevant information.
- The content of the paragraphs has been adjusted in line with recommendations 3, 5, 6, 8, 10, 11, 12, 13, 19, 22, 24and 30, as follows:
- Merge the introductory part, the third paragraph, with the fourth paragraph.
- Merge the introductory part, the first and second paragraphs of point IV.
- Adjust "After four rounds of screening, as shown in Figure 3" to the appropriate position.
- Delete the paragraph "Today, we are in the digital age, where data is ..." as it is repetitive.
- In the background section, merge the first and second paragraphs.
- Merge the paragraph " The most basic cryptographic classification is the division of message encryption into Symmetric Cryptography and Public-Key Cryptography Asymmetric Cryptography." with the following paragraph.
- Merge the two paragraphs of 'Symmetric encryption' in one paragraph.
- Merge the paragraph " The RSA algorithm is described as follows:" with the paragraph above.
- Merge the paragraph " The fourth PQC Standards Conference was held online on November 29, 2022, where aspects of the candidate algorithms were discussed to inform standardization decisions, and submission teams for the selected algorithms, as well as the algorithm submission teams advancing to Round 4, were invited to update their algorithms." with the following paragraph.
- Merge the paragraph " The CCA variant of the Kyber algorithm is …." with the paragraph above.
Thank you very much for your attention and time. Look forward to hearing from you.
Yours sincerely,
Silong Li

Reviewer 2 Report
In this review, the authors first provide a detailed introduction to PQC standardization project carried out by NIST. Afterwards, the Kyber algorithm was emphasized, including the its specific description, software and hardware implementation. At the same time, the authors specifically pointed out the important role played by the PQC algorithm in protecting sensor data from quantum computing attacks. The security of sensor data against quantum computing attacks is an important research Interests, and it is recommended that the authors conduct in-depth discussions in this field.
To my knowledge, NIST has initiated the LWC project and selected the lightweight cryptography standardalgorithms ASCON that tailored for implementation in constrained environments including RFID tags, sensors,and so on. As a post-quantum public key algorithm, Kyber has a different design background from ASCON. If both are used in constrained environment includingsensors, it is recommended to provide a comparison of their functionality and performance.
In addition, there are several minor issues:
(1)The third line of the first paragraph in the main text, 'Johannes A.', should have caused a typo.
(2)In NIST's PQC project, in addition to the four types of algorithms mentioned by the author, there is also a type of supersingular elliptic curve isogeny cryptography that has not been reviewed.
(3)The textual description of the RSA algorithm in the text is inconsistent with the symbols in Figure 4.
(4)In the text below Figure 4 on page 8, the authorspoints out that "the development of post-quantum encryption can be traced back to 1994", but in reality, the emergence of PQC algorithm was much earlier than 1994. In fact, it should be the emergence of Shor algorithm in 1994 that drove the development of PQC algorithm.
(5)For the authors’ information, to resist the attack from a quantum computer, mathematical aporoach of post quantum cryptography introduced here is not the only way. There is another approach named in quantum key distribution (QKD) whose security is independent of computation complexity and hence can in principle resist attacks from any computer. Since the first idea of QKD proposed in 1984 by Bennett and Brassard (BB84 protocol:C.H. Bennett and G. Brassard, in Proceedings of the IEEE International Conference on Computers, Systems, and Signal Processing (Bangalore, India, 1984, p. 175). Moreover, QKD technique has been well developed in the past dozens of years [Advances in Optics and Photonics, 12(4):1012–1236, 2020ï¼›Reviews of Modern Physics, 92(2):025002, 2020] and it is now a practically useful technique rather than a pure theoretical model only. For example, using the practical QKD protocols such as Nature 557, 400 (2018); Phys. Rev. A98, 062323 (2018), QKD has now been demonsteated over very long distances experimentally, including the experiment over 830 km [Nature Photon- ics, 16:1749–4893, February 2022] and the one over 1002 km [Phys. Rev. Lett. 130, 210801 (2023)] .
Author Response
Dear Reviewer:
Thank you for your letter and for the reviewers’ comments concerning our manuscript entitled “Post-quantum Security: Opportunities and Challenges”. Those comments are all valuable and very helpful for revising and improving our paper, as well as the important guiding significance to our researches. We have studied comments carefully and have made correction which we hope meet with approval.
Revised portion are marked in yellow in the paper. The main corrections in the paper and the responds to the reviewer’s comments are as flowing:
- Following the first suggestion, a discussion of post-quantum cryptographic algorithms and sensors has been added in Section IV.
- Add a comparison between LWC and PQC projects to the text as suggested in the second.
- In accordance with recommendation No. 3, remove "Johannes A" in the third line of the text.
- Addition to " supersingular elliptic curve isogeny cryptography" in accordance with recommendation 4
- In accordance with recommendation 5, the logical error in " the development of post-quantum encryption can be traced back to 1994" has been corrected.
- According to the 6th proposal, it is shown in the text that post-quantum encryption algorithms are not the only way to defend against quantum computing attacks.
Thank you very much for your attention and time. Look forward to hearing from you.
Yours sincerely,
Silong Li

Reviewer 3 Report
I have carefully reviewed the proposed work and find it of great interest to the academic community. However, I have observed that the quality of the paper seems fragmented. While some sections are well written and analyzed remarkably, others have very serious flaws that need to be addressed.
Here are some specific points that I believe require attention:
1. At the end of page P2, there seems to be a typographical error where "F'BIO" is mentioned. Please clarify this.
2. I find it difficult to understand why section 2 starts with a paragraph on sensors. It would be better to restructure the section to make it more coherent.
3. The paper should mention also hardware implementation of tightly coupled accelerators, for example 10.1109/ACCESS.2021.3126208.
4. While the history of the algorithms is discussed deeply, some important information is missing. I suggest that section 4 be expanded to include more examples of the state of the art and a comparison of their execution time, hardware complexity, power consumption, and so on. Please differentiate between SW, HW and Mixed HW SW approaches.
5. The paper should be more precise and avoid vague quantifiers and unquoted references such as "Some scientists predict the future based on the current situation..." Either cite a source or provide evidence to support the claim.
6. If sensor security is the primary interest of the paper, I suggest reorganizing the paper structure to make the final aim clear to the reader. As it stands, the paper is too fragmented, particularly in section 5.1 where sensors are discussed again.
I hope these suggestions help improve the quality of the paper.
Author Response
Dear Reviewer:
Thank you for your letter and for the reviewers’ comments concerning our manuscript entitled “Post-quantum Security: Opportunities and Challenges”. Those comments are all valuable and very helpful for revising and improving our paper, as well as the important guiding significance to our researches. We have studied comments carefully and have made correction which we hope meet with approval.
Revised portion are marked in yellow in the paper. The main corrections in the paper and the responds to the reviewer’s comments are as flowing:
- In accordance with the first recommendation, the reference to "F'BIO" has been changed to "Fábio Borges".
- In line with recommendation 2, the sensor-related content in the “background section” has been deleted.
- Added hardware implementation on tightly coupled accelerators in the implementation section, as per recommendation #3.
- In accordance with recommendation 4, the structure of Part IV has been modified and divided into two parts, "Software Implementation" and "Hardware Implementation". At the same time, a brief description of the specific effects achieved is given in each section.
- In accordance with recommendation #5, the paper was revised to try to avoid ambiguous wording.
- In response to recommendation 6, the structure of the paper has been revised by adding a new section 5.1, "Opportunities and challenges", to the "Discussion and evolution" section, and by revising the content of the previous section 5.1 to section 5.2. "evolution".
Thank you very much for your attention and time. Look forward to hearing from you.
Yours sincerely,
Silong Li

Round 2
Reviewer 1 Report
This article can help readers better understand post-quantum encryption techniques and their potential future significance. The article thoroughly understands the Kyber construction principle and the direction that post-quantum encryption will take in the future. you have edited all my comments so thank you.
Author Response
Dear Reviewer:
We feel great thanks for your professional review work on our article.
Thank you very much for your attention. Look forward to hearing from you.
Yours sincerely,
Silong Li
Reviewer 3 Report
I have addressed most of the comments that were raised. However, I still have a few remaining remarks to make.
Regarding point 3, it seems like there might be a reference mismatch issue. There is some confusion between [87] and [89].
As for point 4, I appreciate the distinction between hardware and software. However, I would like to encourage a more detailed comparison of various approaches, along with a summary table.
Author Response
Dear Reviewer:
We feel great thanks for your professional review work on our article. As you are concerned, there are several problems that need to be addressed. According to your nice suggestions, we have made extensive corrections to our previous draft, the detailed corrections are listed below.
First of all, based on your opinion, I rechecked references 87 and 89 and found errors in the citations. At present, the error has been corrected. Second, based on your input, I drew a summary table with information on hardware implementations and software implementations.
Thank you very much for your attention. Look forward to hearing from you.
Yours sincerely,
Silong Li